# Weed Density Detection Method Based on Absolute Feature Corner Points in Field

**Yanlei Xu [1,2], Run He [1], Zongmei Gao [2,*], Chenxiao Li [1,*], Yuting Zhai [1] and Yubin Jiao [3]**

[1] College of Information and Technology, JiLin Agricultural University, Changchun 130118, China; yanleixu@jlau.edu.cn (Y.X.); herun2003@163.com (R.H.); zhaiyutinga@163.com (Y.Z.)

[2] Department of Biological Systems Engineering, Centre for Precision and Automated Agricultural Systems, Washington State University, Prosser, WA 99350, USA

[3] Changchun Institute of Engineering and Technology, Changchun 130117, China; fly-with-wind@163.com

\* Correspondence: zongmei.gao@wsu.edu (Z.G.); licx@jlau.edu.cn (C.L.);
Tel.: +1-509-781-4023 (Z.G.); +86-159-4835-1065 (C.L.)

**Abstract:** Field weeds identification is challenging for precision spraying, i.e., the automation identification of the weeds from the crops. For rapidly obtaining weed distribution in field, this study developed a weed density detection method based on absolute feature corner point (AFCP) algorithm for the first time. For optimizing the AFCP algorithm, image preprocessing was firstly performed through a sub-module processing capable of segmenting and optimizing the field images. The AFCP algorithm improved Harris corner to extract corners of single crop and weed and then sub-absolute corner classifier as well as absolute corner classifier were proposed for absolute corners detection of crop rows. Then, the AFCP algorithm merged absolute corners to identify crop and weed position information. Meanwhile, the weed distribution was obtained based on two weed density parameters (weed pressure and cluster rate). At last, the AFCP algorithm was validated based on the images that were obtained using one typical digital camera mounted on the tractor in field. The results showed that the proposed weed detection method manifested well given its ability to process an image of $2748 \times 576$ pixels using 782 ms as well as its accuracy in identifying weeds reaching 90.3%. Such results indicated that the weed detection method based on AFCP algorithm met the requirements of practical weed management in field, including the real-time images computation processing and accuracy, which provided the theoretical base for the precision spraying operations.

**Keywords:** weed detection; absolute feature corner point; variable spraying; image processing

## 1. Introduction

Field weeds usually grow in inter-row crops, which compete with crops for water, nutrients, and space. Some weeds carry viruses, seriously affecting crop yield and quality [1]. Therefore, herbicides are widely used to prevent weed effects; however, excessive herbicides may cause serious environmental pollution and herbicide waste. To address the issue, weed distribution based variable spraying technology was developed for the precise application rate, with the purpose of achieving minimal farmland pollution and safety of agricultural products, as well as sustainable development of agriculture. Among others, the efficiency and accuracy in weed identification in farmland determine the performance of the spraying technique. The conventional methods for identification of weeds mainly use two attributes between the weeds and crops, i.e., the spatial features and location.

Many studies investigated weed identification methods based on texture, shape, and color. Wajahat et al. [2] used the affine invariant region fusion method for weed detection based on plant leaf shape. Xia CL et al. [3] developed the matching detection of pepper leaves by establishing and training the

pepper leaf shape model to adapt the real leaf image in the greenhouse. Swain K C et al. [4] established a model for the leaf shape in different growth stages of Solanum nigrum and obtained the best fitting through shape comparison to realize the identification of Solanum nigrum. Bakhshipour A et al. [5] applied BP (Back Propagation) neural network to segment wavelet texture features of weed images in sugar beet crops and determined common texture features for multi-resolution images generated by single-stage wavelet transform. Pantazi et al. [6] employed multi-spectral cameras to analyze the spectral characteristics of plants and extracted the seedlings column based on the different spectral characteristics of the collected crops. TD.S.Guru et al. [7] classified different flowers by the color texture matrix, gray level co-occurrence matrix, and windowed Fourier transform of the flower image. Dongjian He et al. [8] fused three features of common weeds in corn fields in the GuanZhong area including leaf shapes, textures and fractal dimensions, and realized weed identification based on multi-feature fusion. All these studies focused on single plant or small-scale plants recognition; with high accuracy in identification, the integrated algorithms are relatively complex. Therefore, real-time image processing is difficult to realize. In order to meet the real-time requirement, low-resolution images are often used as input data. In the processing of variable spraying and machine navigation, visible cameras are mounted on top of a machine, i.e., tractor or vehicle, with a certain angle adjusted according to the actual situation, aiming for imaging multi-row of plants; high resolution images are required. Thus, traditional weed identification methods are limited in real-time for processing high-resolution images.

During the crop growth, weeds among individual plants and inter-row weeds have different effects on crops. Weeds among individual plants slowly lose their competitiveness and invisibility with the growth of crops, while weeds located inter-row growing well have adverse effects on crops. Generally, crops are planted in fixed row space, while inter-row weeds are located between crop rows and can be distinguished according to the position. The position histogram and Hough transformation are two methods commonly used in position-identification. Wenhua Mao et al. [9] used the pixel position histogram to identify crop rows according to the position characteristics of row crops and then filled and eliminated the connected area of wheat crops using the seed filling method, obtaining the weed area. The position histogram method is user-friendly, yet with large relative error and accuracy being around 80%, which may cause low accuracy in the subsequent weed density detection, especially for complex (illumination, overlapping, weather, and so on) farmland in real-time farmland work. Therefore, the position histogram is not appropriate for real-time field work. Astrand et al. [10] used Hough transformation to divide crop ridges into regular rectangles to obtain inter-row crop areas. Zhihua Diao et al. [11] extracted crop line centerline by deviation analysis of relative distance, combining random Hough transformation with transformation from image coordinate system to the world coordinate system. Qin Zhang et al. [12] extracted rice grayscale features in the S component based on the HSI (hue, saturation, value) color model. The feature points were clustered using the nearest neighbor method, and the rice row center line was extracted by the known point Hough transformation. Hough transformation could detect the crop row position accurately. However, Hough transformation is not ideal for detecting incomplete lines and curves, and the calculation burden is relatively heavy. The time complexity of Hough transformation is $O(N^2M)$; $M$ and $N$ are the size of image. When processing multiple ridges at the same time, it is difficult to meet the real-time spraying requirements. Qingkuan Meng et al. [13] segmented two-dimensional image using K-means clustering method and determined the crop row position using particle swarm optimization algorithm according to the characteristics of crop rows. Although the particle swarm optimization algorithm has a higher accuracy rate for crop line detection, the real-time performance is not significantly improved compared to the Hough transform and it is unable to meet the high-efficiency variable spraying work demand in real time. Shikai Huang et al. [14] grayed the farmland color image to reduce the effect of natural light using the Q component in the YIQ (National Television Standards Committee) color space. The center line was identified using Hough transformation, and crop area was marked on both sides of the center line by measuring corn leaf width and using world coordinate system transformation. This method

could be applied to different illumination conditions, yet the operation was inconvenient for different field applications, due to the manual measurement of actual crop row width. Further, Hough transform was used to identify centerline, leading to poor real-time performance.

For improving the real-time performance of weed identification, this study proposed a rapid method to detect weed density based on absolute feature corner point (AFCP) algorithm. The specific objectives of this study were:

(1) To propose a robust weed density identification method comprised of three modules: image preprocessing, crop row detection and weed density detection.
(2) To develop an AFCP algorithm including sub-corner classifier and an absolute corner classifier capable of detecting crop row and the weed position.
(3) To calculate the weed pressure and weed cluster rate.
(4) To validate the proposed method based on AFPC algorithm by applying herbicide on weeds distributed within a corn field.

## 2. Materials and Methods

The weed density detection method included three modules: image preprocessing, crop row detection, and weed density detection, as shown in Figure 1. Image preprocessing was composed of image acquisition, optimal region extraction, image grayscale, threshold binarization segmentation, and morphological optimization. Crop row detection included four parts: connected area label, sub-corner point classification, absolute corner point location, and crop row identification. Furthermore, weed density detection included weed region extraction, world coordinate system transformation, and weed density parameter calculation.

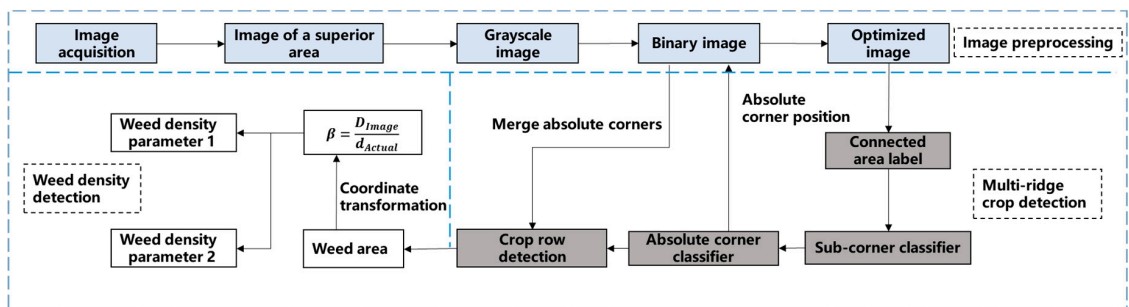

**Figure 1.** Block diagram of weed density detection proposed in this study.

### 2.1. Image Preprocessing

#### 2.1.1. Image Acquisition and Optimal Region Extraction

The Meizu MX5 camera (Sony IMX220 lens, Sony, Tokyo, Japan) was used to obtain farmland images in the experimental field of JiLin Agricultural University, ChangChun, China. The camera has a back-illuminated CMOS sensor, with a resolution of $5248 \times 3936$ pixels. The horizontal vertical resolution is 72 dpi, and the effective pixel is 20.7 million. The camera was mounted on top of a tractor, 250 cm away from the ground and 40° angle to the ground. Matlab R2017a (The Mathworks, Natick, MA, USA) was used as software platform to process the farmland images. The computer was configured with Intel Core (TM) i5, 3.1 GHz, and 4 G memory.

Figure 2 was a corn farmland image at seedling stage with weeds obtained in the experimental field of JiLin Agricultural University on 3 June 2019. The weeds were labeled using the blue circles. It can be observed that weeds were small and mostly distributed among rows. As mentioned above, the camera was mounted with 40° angle to the ground, resulting in distant crop row existing distortion. In order to improve the correct recognition rate and reduce the complexity of the algorithm, the images were divided into three parts equally from bottom to top according to the position, i.e., optimal,

sub-optimal, and other areas. The subsequent simulation and field experiments showed that weed distribution could be detected quickly and accurately only by processing the optimal area.

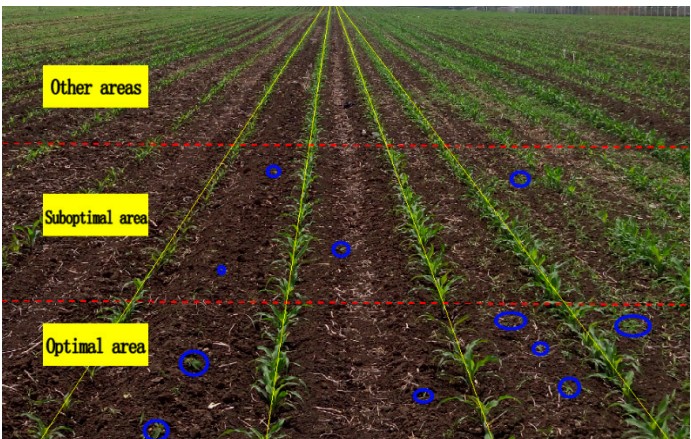

**Figure 2.** Typical image acquired from corn at seedling stage with weeds (blue circle) obtained in the experimental field. The image was segmented into three areas: optimal, suboptimal, and other areas.

### 2.1.2. Graying and Smoothing

The crop and the background could be segmented based on the color feature, with significant color differences between them. The improved particle swarm optimization (IPSO) algorithm [15] was used to extract the optimal color feature coefficients, which were used to gray the farmland images. The reverse mutation strategy can reduce the interference of external environment and improve the accuracy of global search, as well as avoid the optimal particle falling into local optimum. On-line and off-line processing were combined to ensure the rapidity and accuracy of the algorithm. Optimal color coefficient extraction was executed off-line.

After gray-scale processing, the image still contained noise. In order to retain image details, extremum median filter was adopted to suppress the noise of the image [16]. This method can effectively eliminate random noise caused by weather or complex circumstances. Compared with the mean filtering method, the extremum median filtering can achieve improved performance for retaining image details and effectively suppress the noise while ensuring image clarity.

The extreme median filtering for a grayscale image is implemented as follows:

The matrix $X = \left[x_{ij}\right]$ represents a digital image ($i, j$ represents the pixel position), $W_n\left[x_{ij}\right]$ represents to process image on the neighborhood window of $n \times n$ ($n$ is odd integer), centered on point ($i, j$). Matrix $Y = \left[y_{ij}\right]$ represents the output image, and $Med\left(W_n\left[x_{ij}\right]\right)$ represents the median value of all pixels in the window $W_n\left[x_{ij}\right]$. Equation (1) determines whether pixel is a noise point, and Equation (2) is a filtering method.

$$x_{ij} \in \begin{cases} Noise \ x_{ij} = \min\left(Wn\left[x_{ij}\right]\right) \ or \ x_{ij} = \max\left(W_n\left[x_{ij}\right]\right) \\ Signal \ \min\left(W_n\left[x_{ij}\right]\right) < x_{ij} < \max\left(W_n\left[x_{ij}\right]\right) \end{cases} \tag{1}$$

$$y_{ij} = \begin{cases} Med\left(W_n\left[x_{ij}\right]\right) x_{ij} \in Noise \\ x_{ij} \ x_{ij} \in Signal \end{cases} \tag{2}$$

### 2.1.3. OTSU (Maximum between-Cluster Variance) Threshold Image Segmentation

After extreme value median filtering processing, the crop row and weed in the image were sufficiently clear for the analysis. In order to further extract the effective information of weed distribution, the OTSU threshold (maximum between-cluster variance) method [17] was used to convert the grayscale image into binary image.

The threshold value $t$ is very important to segment image to foreground and background. Let $T$ represent the gray level of an image. According to threshold value $t$, image pixels can be divided into two categories. The probability is calculated when pixel gray value equals to $i$. Furthermore, the probability $F_0$ of background image with gray value less than or equal to $t$ can be obtained. Similarly, the probability $F_1$ of foreground image can be obtained from Equations (3) and (4).

$$F_0 = \sum_{i=0}^{t} p_i \tag{3}$$

$$F_1 = 1 - F_0 = \sum_{i=t+1}^{T-1} p_i \tag{4}$$

Then the background and foreground gray mean values $\mu_0$ and $\mu_1$ are shown in Formulas (5) and (6).

$$\mu_0 = \sum_{i=0}^{t} i\frac{p_i}{F_0} \tag{5}$$

$$\mu_1 = \sum_{i=t+1}^{T-1} i\frac{p_i}{F_1} \tag{6}$$

The inter-class variance between foreground and background is:

$$\sigma^2 = F_0\left(\mu_0 - \sum_{i=0}^{T-1} ip_i\right)^2 + F_1\left(\mu_1 - \sum_{i=0}^{T-1} ip_i\right) \tag{7}$$

When inter-class variance is largest, the value $t$ is the optimal threshold, as shown in Formula (8).

$$t = \mathrm{argmax}\{\sigma^2(t)\}, 0 \le t \le T - 1 \tag{8}$$

### 2.1.4. Image Optimization

In order to make crop row more prominent and obtain the position information with higher accuracy, the smaller weeds were removed using morphological optimization with $3 \times 3$ square structure element. Weed identification would be carried out after the recognition of crop rows so the image before optimization was used as the initial picture. Such method can reduce the computation time of the subsequent algorithm, with improving the accuracy and reducing the impact of small weeds on the crops.

Mathematical morphology is a nonlinear filtering method, which generally includes erosion, dilation, open operation, and close operation [18–20]. Mathematical morphology uses mathematical tools to process image structure components. It has good effects in reducing noise, image edge feature extraction, and image restoration and reconstruction.

Firstly, appropriate optimization factors are selected to erode and dilate the binary image. Secondly, the dilation of binary image $bw(x, y)$ by optimizing factor $S_{tc}(x', y')$ is denoted as $S_{tc} \oplus bw$, and defined as:

$$\left(S_{tc} \oplus bw\right)(x, y) = \max\{bw(x - x', y - y') + S_{tc}(x', y') | (x', y') \in Z_{S_{tc}}\} \tag{9}$$

Erosion operation is denoted as $Stc \odot bw$, and defined as:

$$\left(S_{tc} \odot bw\right)(x, y) = \min\{bw(x + x', y + y') + S_{tc}(x', y') | (x', y') \in Z_{S_{tc}}\} \tag{10}$$

where *bw* is the input binary image, $S_{tc}$ is the optimization factor, $Z_{Stc}$ is the definition domain of $S_{tc}$, and *bw*(*x*,*y*) is assumed to be −∞ outside the definition domain.

### 2.2. Multi-Ridge Crop Row Detection

At present, the straight detection based on Hough transform is the most popular method in multi-ridge crop row detection. This method performs well at detecting the crops with small curvature but the Hough transform has a large amount of calculation and poor real-time performance, especially when it is used for multi-row crop detection. The row crops are composed of a single crop, which can be detected by recognizing the top and bottom plants of row crops. In the study [21], authors found that Harris corner detection algorithm is favorable for the crop row detection. This algorithm not only detects all corners of curvilinear elements like crops and grass but also meets the real-time requirement.

#### 2.2.1. Harris Corner Detection

Harris corner detection is proposed by C. Harris on the basis of Moravec algorithm. It is based on image gray information and calculates edge curvature and gradient to identify the corners [22]. The Harris corner detection is widely used in image registration, image stitching, image merging, precise positioning, and other fields [23–27]. It can greatly reduce the calculation burden and the noise which could be able to adapt the gray transformation and overlap. Harris corner detection method is accurate for detecting corners in curves [28]. The procedure for operating Harris corner detection is implemented as follows:

We construct a rectangular window centered on the coordinates (*x*, *y*) of a pixel in the binary image which can move in any direction. Let rectangular window move *u* in the *X*-axis direction and *v* in the *Y*-axis direction. Based on Taylor expansion, the grayscale change is expressed by analytical Formula (11) [29–32]:

$$E(x,y) = \sum W_{x,y}\left(I_{x+u,y+v} - I_{x,y}\right)^2 = \sum W_{x,y}\left(u\frac{\partial I}{\partial X} + v\frac{\partial I}{\partial y} + o\left(\sqrt{u^2+v^2}\right)\right)^2 \tag{11}$$

where *E* is the grayscale variation in the window, *W* is the window function, *I* is the image grayscale function. Generally $w_{x,y}$ is defined as:

$$W_{x,y} = e^{-(x^2+y^2)/\delta^2} \tag{12}$$

$$\text{So}: E(x,y) = \sum e^{-(x^2+y^2)/\delta^2}\left[u^2(I_x)^2 + v^2\left(I_y\right)^2 + 2uvI_xI_y\right] = Au^2 + 2Cuv + Bv^2 \tag{13}$$

where:

$$A = (I_x)^2 \bigotimes w_{x,y}, B = \left(I_y\right)^2 \bigotimes w_{x,y}, C = \left(I_x{\cdot}I_y\right)^2 \bigotimes w_{x,y}\left(\bigotimes \text{means convolution}\right) \tag{14}$$

Convert *E*(*x*, *y*) to quadratic:

$$E(x,y) = [u\ v]M\begin{bmatrix} u \\ v \end{bmatrix} \tag{15}$$

where *M* is the real symmetric matrix:

$$M = W_{x,y}\begin{bmatrix} I_x^2 & I_xI_y \\ I_xI_y & I_y^2 \end{bmatrix} \tag{16}$$

$I_x$ is the gradient in the *x* direction of image *I*, and $I_y$ is the gradient in the *y* direction of image *I*. By analyzing the matrix *M*, the eigenvalue of *M* can be obtained. If the eigenvalue value is big, it indicates that the point is a corner point, and the corner response function *CRF* is defined as:

$$CRF = det(M) - k{\cdot}trace^2(M) \tag{17}$$

where *det* is the determinant of the matrix *M*, *trace* is the trace of the matrix *M*, *k* is usually a constant 0.04. If *k* is bigger, less false corners and more real corners would be obtained. If *k* is smaller, more corners would be acquired; meanwhile, some true corners would be missed. The point located at the local maximum of *CRF* is considered as the corner.

### 2.2.2. Redundant Corners Optimization

The number of corners detected by Harris method was big. In order to reduce the influence on algorithm caused by redundant corners, Harris method was optimized, and threshold parameters were set to greatly reduce redundant corners of the same attribute and the calculation burden of the algorithm. The specific operation is as follows.

The absolute difference between the gray value of the center pixel and the gray value of any pixel in the neighborhood was set as $\Delta$, the threshold value was as *t*. If $\Delta \leq t$, central pixel point was similar to the adjacent pixel point (where threshold value *t* depended on the gray level, and it was required to describe crops and weeds as comprehensively as possible). For $3 \times 3$ detection window (this size was most appropriate by experiments), eight adjacent points were compared with central point, and the number of points that correspond to $\Delta \leq t$ was:

$$E(i, j) = \sum_{x,y} F(i + x, j + y)|x, y = \{-1, 1\} \tag{18}$$

where:

$$F(i + x, j + y) = \begin{cases} 1, \Delta(i + x, j + y) \leq t \\ 0, \Delta(i + x, j + y) > t \end{cases} \tag{19}$$

So, the value range of $E(i, j)$ was from 0 to 8. When $E = 0$, it indicated that there was no similarity between center point and adjacent point. The center point was highly likely an isolated noise point and thereby was discarded. When $E = 7$ or 8, it indicated that center point must be the internal point of image pixel, which did not meet the requirements of corners and was discarded as well. When $E = (1-6)$, center point had $E$ (1–6) similarity points with its adjacent points, which most likely were corners and reserved. $E = 3$ was taken as an example, the relationships between 8 adjacent points are shown in Figure 3.

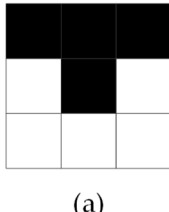 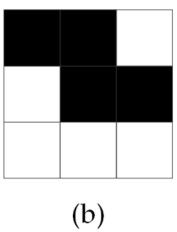 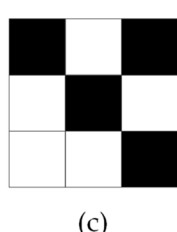

(a)　　　　　　　　　　(b)　　　　　　　　　　(c)

**Figure 3.** The number of corners detected by improved Harris method with pixel adjacent point when $E = 3$: (**a**) 3 adjacent point; (**b**) 2 adjacent point; (**c**) 1 adjacent point.

For Figure 3a, which contained three adjacent points (black square), any one of the three points was taken as the candidate corner due to the same type. For two adjacent points in Figure 3b, any one single adjacent point was taken as candidate corner. There were no adjacent points in Figure 3c, so all points were reserved. Similarly, when $E = \{4, 5, 6\}$, adjacent corners of the same type can be filtered out to remove redundant corners.

### 2.2.3. Crop Row Corner Detection

In order to detect absolute corners of each crop row more accurately and quickly and to eliminate pseudo-absolute corners of weeds, two absolute corner classifiers: sub-absolute corner detector and absolute corner detector were proposed and applied in this study. The two classifiers can discriminate

the corners at the edge of crops and weeds and can detect the absolute corners with absolute position features. The area of crop row can be obtained, and multi-row crop detection can be realized by connecting absolute corners. The absolute corners were the corners at four absolute positions of each crop row, as shown in Figure 4.

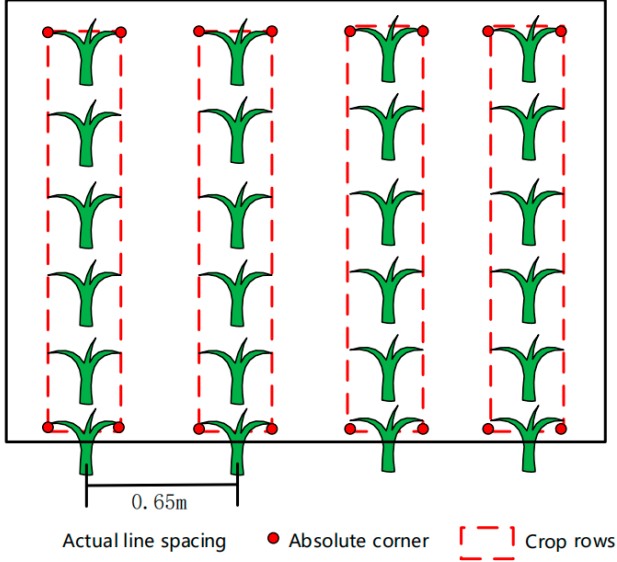

**Figure 4.** Schematic diagram of absolute corner.

Sub-Absolute Corner Detector

The crop row consisted of individual plant, and the individual plant was considered as a sub-connected region. A minimum bounding box (MBB) was used to process the sub-connected region, as shown in Figure 5. The corners in the sub-connected region were called sub-corners (SCO). The red dotted box was the MBB of the connected area; four vertices of the MBB were called the sub-contrast corners (SCC) of the crop row. The centroid of the sub-connected region was called the sub-centroid (SCE).

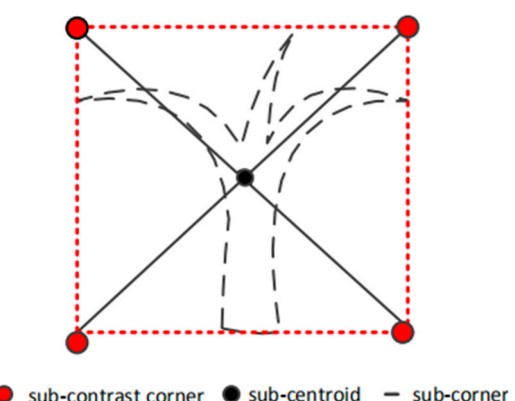

**Figure 5.** Sub-corner point diagram.

Scanning on each sub-connected region, the sub-contrast corner, the sub-corner, and the sub-centroid of each sub-connected area can be obtained. The method of MBB scanning was used to scan each sub-enclosing rectangle, and the sub-absolute corners (SAC) were obtained. The detection process was as follows:

(1)   Set the sub-absolute corner scanning window as $w_1 \times h_1$. The window function was the MBB of the connected region, and $w_1$ and $h_1$ were the width and height of the MBB.

(2) Set the horizontal (*x*-axis) and vertical (*y*-axis) coordinate component difference functions. Set the point coordinates of different type as: SCC point *Con(x,y)*, SCO point *Cc(x,y)*, SCE point *Oc(x,y)*, SAC point *Ca(x,y)*. The horizontal coordinate component difference function *d(x)* and the vertical coordinate component difference function *d(y)* of any SCO and SCC were described as:

$$
\begin{cases}
d(x) = x_{Cc} - x_{Con} \\
d(y) = y_{Cc} - x_{Con}
\end{cases}
\tag{20}
$$

where $x_{Cc}$ and $y_{Cc}$ were abscissa and ordinate of SCO *Cc(x,y)*, $x_{Con}$ and $y_{Con}$ were abscissa and ordinate of SCC point *Con(x,y)*.

(3) Set the classifier. Assuming that the abscissa $x_{Cc}$ of the SCO point *Cc* was the candidate corner point of SAC, the abscissa $x_{Ca}$ of SAC was horizontal coordinate when the horizontal component difference between SCO point and SCC was minimum. Similarly, the ordinate $y_{Ca}$ was vertical coordinate when the difference between the vertical components was the minimum:

$$
x_{Cc} \in x_{Ca} = \mathrm{argmin}\{|d(x)|\}, ul[x_{Con}] \leq x \leq ur[x_{Con}]
\tag{21}
$$

$$
y_{Cc} \in y_{Ca} = \mathrm{argmin}\{|d(y)|\}, ul[y_{Con}] \leq y \leq ll[y_{Con}]
\tag{22}
$$

where, $x_{Ca}$ and $y_{Ca}$ were abscissa and ordinate of SAC point *Ca(x,y)*. *ul* represented upper left position. *ul*[$x_{Con}$] represented abscissa of SCC point *Con(x,y)* in upper left position. Similarly, *ur* represented upper right position. *ur*[$x_{Con}$] represented abscissa in upper right position. *ul*[$y_{Con}$] represented ordinate in upper left position. *ll* represented lower left position. *ll*[$y_{Con}$] represented ordinate in lower left position.

(4) The relative position classification of the corner points in accordance with step 3 was carried out, and the four sub-contrast corner points were classified into four categories according to the position:

$$
ul[Ca(xy)]: \ C_{c(x,y)} \in ul, \ d(x) \geq 0 \ \& \ d(y) \geq 0
$$

$$
ll[Ca(xy)]: \ C_{c(x,y)} \in ll, \ d(x) \geq 0 \ \& \ d(y) \geq 0
$$

$$
ur[Ca(xy)]: \ C_{c(x,y)} \in ur, \ d(x) \geq 0 \ \& \ d(y) \geq 0
$$

$$
lr[Ca(xy)]: \ C_{c(x,y)} \in lr, \ d(x) \geq 0 \ \& \ d(y) \geq 0
\tag{23}
$$

where *ul* [*Ca(x,y)*] was SAC located in upper left position. *ll*[*Ca(x,y)*], *ur*[*Ca(x,y)*] and *lr*[*Ca(x,y)*] were SAC in lower left, upper right, and lower right position respectively.

(5) Repeat step 3 and 4 until scanning the entire image.

Absolute Corner Detector

Using the sub-absolute corner classifier, SAC *Ca(x,y)* with positional attributes (*ul*, *ll*, *ur*, *lr*) and SCE *Oc(x,y)* were obtained. SAC and SCE contained the position information of each plant. It was necessary to scan and identify the SCE and find the absolute centroid (AE) *O(x,y)* and absolute corner (AC) *C(x,y)* of each row, which can accurately detect crop line information. Considering time complexity of the algorithm, the scanning window nearest neighbor method was used to classify the SCE according to the spatial distance relation, which can classify crop rows. However, some inter-row weeds were classified as pseudo-crop rows. It has been found through experiments that the angle between AC and AE of the pseudo-crop row was significantly different from that between AC and AE of the crop row. Setting the angle threshold value can eliminate pseudo-absolute corner points. $\theta$ was the interior angle $\angle C_{ul}OC_{ur}$ of triangle formed by $C_{ur}$, $C_{ul}$, and *O*, where $C_{ur}$ was the AC located in upper right position, $C_{ul}$ was the AC located in upper left position and *O* was the AE. And, $\angle C_{ul}OC_{ur} = \angle C_{ll}OC_{lr}$ where $C_{ll}$ was the AC located in lower left position, $C_{lr}$ was the AC located in lower right position. The algorithm flow was as follows:

(1) Set the distance function D of the SCE, and set $(x_{oc1}, y_{oc1})$ and $(x_{oc2}, y_{oc2})$ as the SCE to evaluate, then:

$$D = \sqrt{(x_{oc1} - x_{oc2})^2 + (y_{oc1} - y_{oc2})^2} \tag{24}$$

(2) Calculate average distance threshold $\overline{D} = \frac{\sum_{k=1}^{n-1} D_{k,k+1}}{n-1}$, where $D_{k,k+1}$ was the distance between two adjacent sub-centroids, and $n$ was the total number of SCE.

(3) Perform $j$ times scans and pre-sorting. Search the closest centroid for each SCE in scanning process. If their distance was less than the average distance threshold $\overline{D}$, then they belonged to one class, otherwise a new class appeared.

(4) Repeat step 2 and 3 until all sub-centroids in the image were classified and the centroid classes were obtained.

(5) Take the absolute corners of each centroid class. The selection method of absolute corners (ACs) was as follows:

The vertical distances of all sub-centroids in the class were calculated on the axis $y = 0$ and the line $y = H$. Two sub-centroids with the minimum distance were taken. The four absolute corners as candidates were closest to axis $y = 0$ and closest to straight line $y = H$.

(6) Remove the pseudo absolute corners. Taking $y = 1/2 \, H$ as the horizontal central axis, sub-centroid with the smallest Euclidean distance to the line $y = 1/2 \, H$ was the absolute centroid. Set the angle $\theta$ threshold as $\theta \in \left[23°\text{–}47°\right]$. If the angle value between candidate absolute corner and the absolute centroid was in the angle threshold, the candidate absolute corner was deemed to be absolute corner; otherwise, it was discarded and the searching continued.

(7) Repeat step 6 until the absolute corners for all classes were found.

After the absolute corners were obtained, the crop row can be detected by merging the ACs in each centroid class.

### 2.2.4. Crop Row Detection

The specific method for merging absolute corners was detailed as follows. The quadrangle formed by connecting the four absolute corners belonging to the same centroid class was a crop row, as shown in Figure 6. Figure 6 showed four absolute corners belonging to the same centroid class which were connected and marked. The connected area was considered to be a crop row.

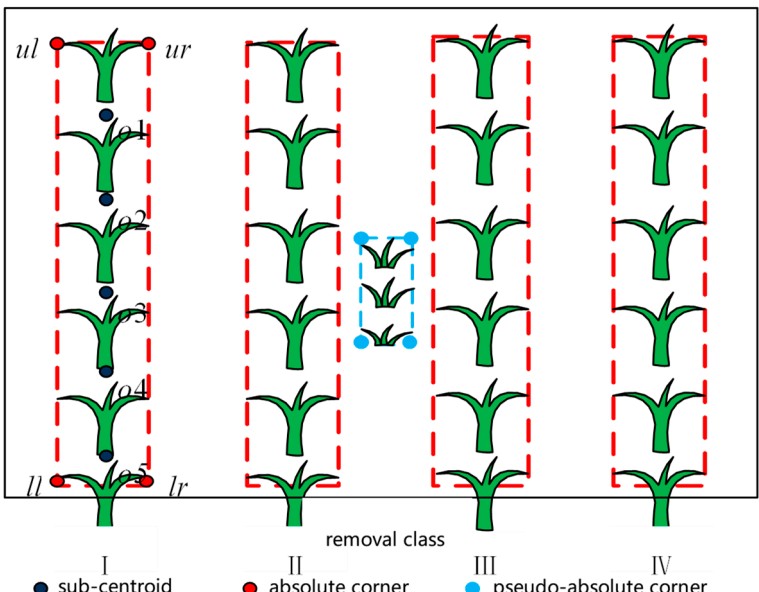

**Figure 6.** Schematic diagram of crop row detection based on merging absolute corners.

### 2.3. Weed Density Detection

#### 2.3.1. Crop Line Elimination

The crop row position information extracted in Section 2.2.4 was obtained from the binary image without optimized processing. All pixels in crop row were set to 0, and the remaining white pixels were inter-row weeds. The specific formula was as follows:

$$\left[M_x, N_y\right] = find(Clear == 1) \tag{25}$$

where the matrix $M_x$ and $N_y$ stored the $x$-coordinate and $y$-coordinate information of weeds, respectively, *Clear* was the binary image eliminating the crop lines. The number of white pixels was counted after deleting the crop rows, which was the number of weed pixels.

#### 2.3.2. Weed Density Calculation

Generally, to calculate the actual weed distribute density, reference object was set in the field and world coordinate system of image pixel and actual area was obtained by processing and identifying the reference object image. However, this method was redundant. In most regions of JiLin province, the row pitch of corn was fixed. Using absolute corner points, the row pitch of crop rows can be obtained. The world coordinate system coefficient can be obtained by the ratio between the row pitch in the image and the actual fixed row pitch, which can detect the row width in real time and improve the recognition accuracy. Instead of measuring crop row pitch manually in the traditional method, this method can eliminate the cumbersome steps of pre-operation measurement.

Currently, weed pressure and cluster rate were used to describe the weed density [33]. In this study, two density parameters of weed pressure and cluster rate were set, in which weed pressure parameter was the ratio of the weed quantity to the crop quantity and cluster rate parameter was the ratio of the weed quantity to the land area.

Weed pressure parameter $\varphi_1$ was:

$$\varphi_1 = \frac{p_0}{p_1} \times 100\% \tag{26}$$

Weed cluster rate parameter $\varphi_2$ was:

$$\varphi_2 = \frac{p_0}{\beta^2 \times S_{actual}} \times 10 \tag{27}$$

where $p_0$ was the number of weed pixels in the optimal region, $p_1$ was the number of crop pixels in the optimal region, $\beta$ was the world coordinate system coefficient, and $S_{actual}$ was the actual area of the optimal region.

For getting the four absolute corners of the same crop row $C_{ul}(x, y), C_{ur}(x, y), C_{ll}(x, y), C_{lr}(x, y)$, the center horizontal coordinate $\frac{x_{lr}-x_{ll}}{2}$ of the crop row was calculated, the distance between the two adjacent center coordinates was $s$. The average distance was $\bar{s} = \frac{\sum_{j=1}^{i-1} S_j}{i-1}$, where $i$ was the number of crop rows. The fixed row pitch of corn in JiLin area was 0.65 m, then the world coordinate system parameter $\beta = \frac{\bar{s}}{0.65}$ can be obtained. The calculation formulas of row width $L_{width}$ and line length $L_{lenth}$ of crop lines in the image were as follows:

$$L_{width} = x_{rb} - x_{lb} \tag{28}$$

$$L_{length} = \sqrt{\left(\frac{x_{ru} - x_{lu}}{2} - \frac{x_{rb} - x_{lb}}{2}\right)^2 + \left(\frac{y_{ru} - y_{lu}}{2} - \frac{y_{rb} - y_{lb}}{2}\right)^2} \tag{29}$$

The actual width and length of the crop row ($L_{aw}$,$L_{al}$) were calculated as:

$$L_{aw} = \frac{L_{width}}{\beta}, \ L_{al} = \frac{L_{length}}{\beta} \tag{30}$$

The area conversion formula was:

$$\beta^2 = \frac{S_{image}}{L_{aw} \times L_{al}} \tag{31}$$

According to the above steps, the real-time world coordinate system in the field can be calculated. The weed cluster rate in the unit land area and pressure parameters of weeds on crops can be obtained, which provided a theoretical basis for establishing the field weed models.

### 2.4. Field Validation

For evaluating the accuracy and rapidity of the method proposed by this study, a field spraying experiment was carried out in JiLin agricultural experimental field on 3 June 2019 under natural light conditions. Maize plant spacing was 0.2 m, row spacing was 0.65 m, and the high-ground spray rod sprayer was selected for the experiment. Cameras were mounted on the machine and the distance between the camera and the ground was 2.5 m. The view angle was 40° and the camera frame rate was set as 20 frames/s, as shown in Figure 7a. To achieve precise variable spraying [34], the target six nozzles were set as two groups, and the solenoid valve was set before the nozzle tube, as shown in Figure 7b. The Figure 7c was the spaying machine for farmland work. The computer extracted the image every 15 frames. The obtained images were processed using the method proposed earlier. Then the weed distribution density can be obtained. According to different weed distribution density, the electromagnetic valves opened or closed, which controlled the nozzle spraying status. In the experiment, the real-time weed identification accuracy and the spraying results were tested.

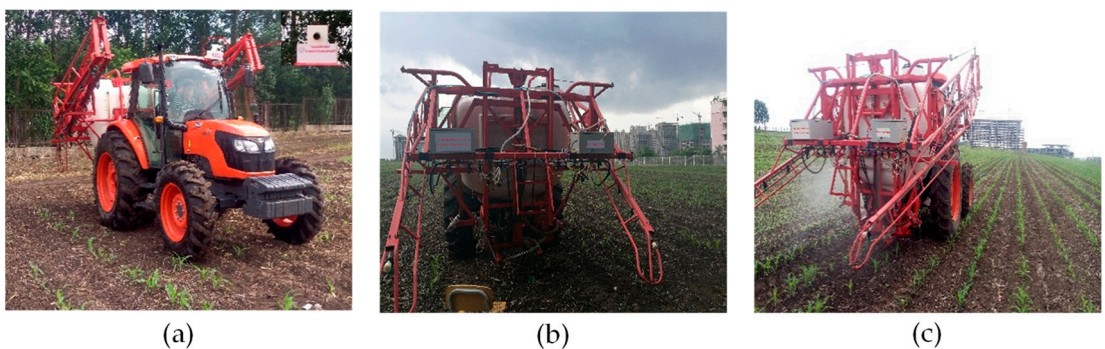

| (a) | (b) | (c) |

**Figure 7.** Field validation experiment using the proposed method: (**a**) Camera placement; (**b**) Image recognition for target nozzle; (**c**) Continuous spray.

## 3. Experiment Results

### 3.1. Image Preprocessing Experiment

The optimal area of the collected image (shown in Figure 8a) was grayed by RM-PSO algorithm and the grayscale image is shown in Figure 8b. Figure 8c,d shows the result images using mean filtering and extreme median filtering, respectively. It was obvious that the extreme median filter in Figure 8d could effectively retain image details and suppress noise.

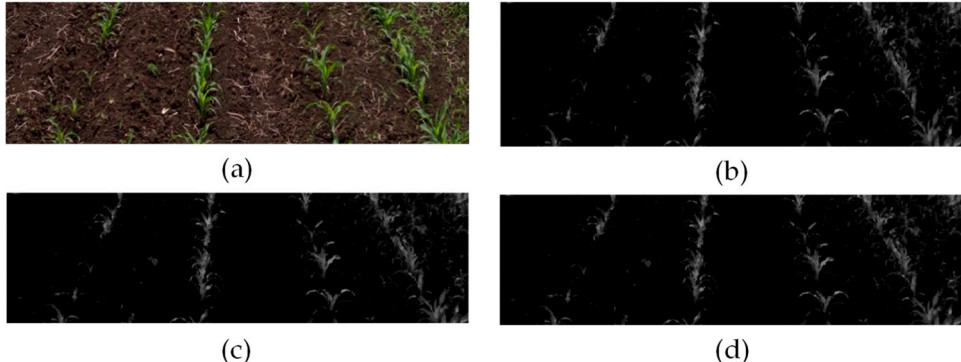

**Figure 8.** Preprocessing of image acquired in field: (**a**) Color image of farmland; (**b**) Grayscale image; (**c**) Mean filter image; (**d**) Extreme median filter image.

After grayscale processing, the image was segmented into binary images by OTSU threshold method and the result image is shown in Figure 9a.

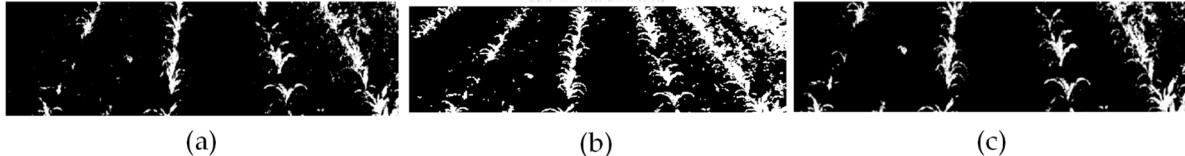

**Figure 9.** Segmentation and optimization of the preprocessed image. (**a**) Binary image segmentation; (**b**) Morphological optimization image; (**c**) Tiny area elimination.

In processing of optimization, the appropriate optimization factors were very important. By comparing different dilation and erosion parameters, it was found that the dilation parameter 2 and the erosion parameter 1 were optimal parameters. The optimization image using the optimal parameters is shown in Figure 9b. For reducing the influence of tiny weeds on subsequent processing, areas with pixel values less than 15 pixels were removed, as shown in Figure 9c.

*3.2. Crop Row Corner Detection*

The MBB was used to process the connected region in the optimized image, as shown in Figure 10. Secondly, Harris method was used to extract the corner information of each sub-connected region, as shown in Figure 11.

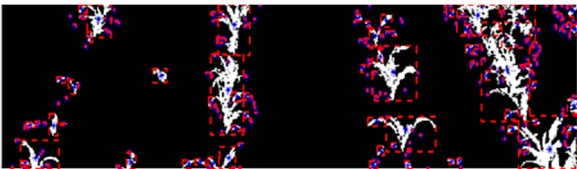

**Figure 10.** Sub-connected regions obtained from Figure 9c.

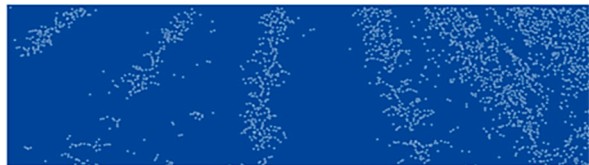

**Figure 11.** Extraction of corners of crop and weed by Harris.

Figure 12a shows a binary map of a single crop obtained from a crop row and Figure 12b shows the sub-corner information of the individual plant. Figure 12c shows the corresponding position

information of the detected sub-corners for the individual plant. Figure 12d shows the corresponding position information of sub-corners that removed redundancy.

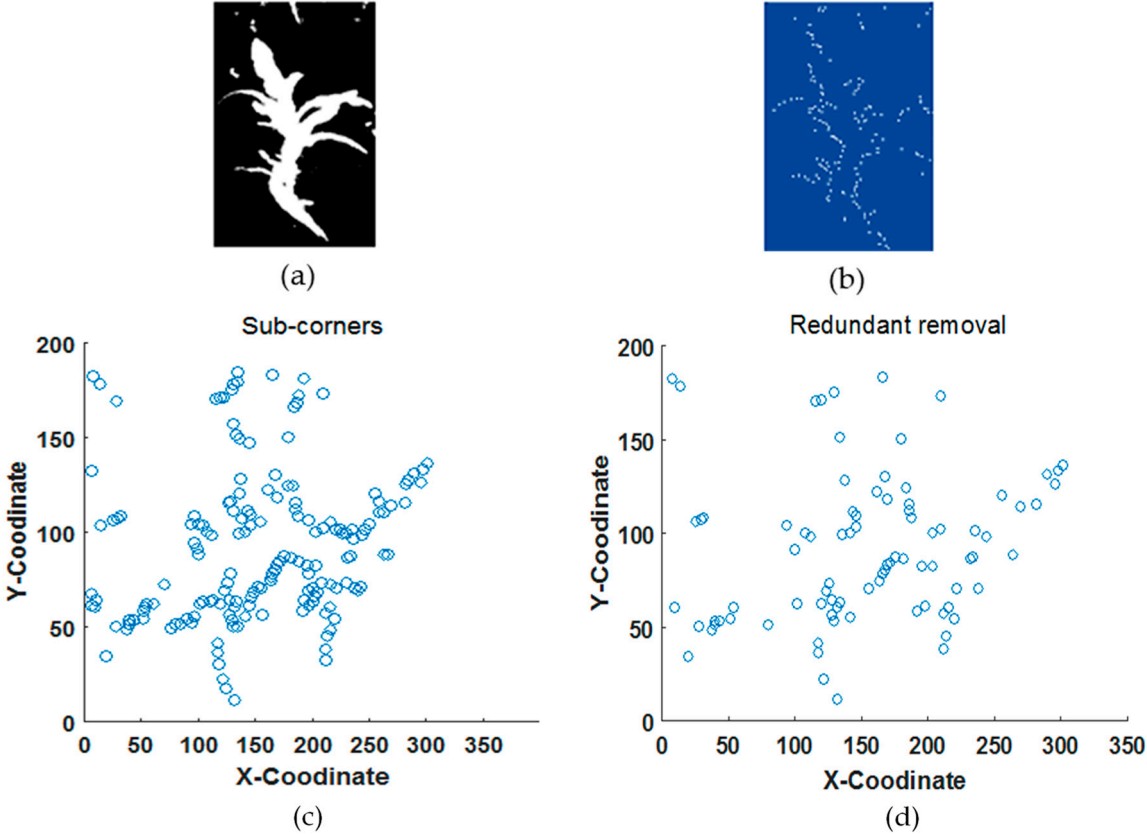

**Figure 12.** Corner information of individual crop: (**a**) Individual crop; (**b**) Individual crop sub-corner; (**c**) Sub-Corner position information; (**d**) Redundant removal of sub-corner position information.

The above sub-corner information was processed by the sub-absolute corner detector proposed in Section 2.2.3, with which the sub-absolute corner information can be obtained. Then, the sub-absolute corner was processed by the absolute corner detector proposed in Section 2.2.3; the obtained absolute corner information is shown in Table 1.

There were pseudo absolute corners among the above absolute corners. In order to classify the absolute corner, eliminating the pseudo absolute corner was necessary. One hundred crop and 100 weed angles between absolute corners and absolute centroid which were extracted from 200 farmland images were calculated, respectively. The statistics results are shown in Figure 13.

The $\theta'$ of the pseudo-crop row class was always greater than the $\theta$ of the crop row class. Crop row type angle threshold was at $\left[23^\circ \sim 47^\circ\right]$ and weed class angle threshold was at $\left[68^\circ \sim 180^\circ\right]$. Therefore, the range of crop row threshold was set to eliminate the pseudo-crop row class.

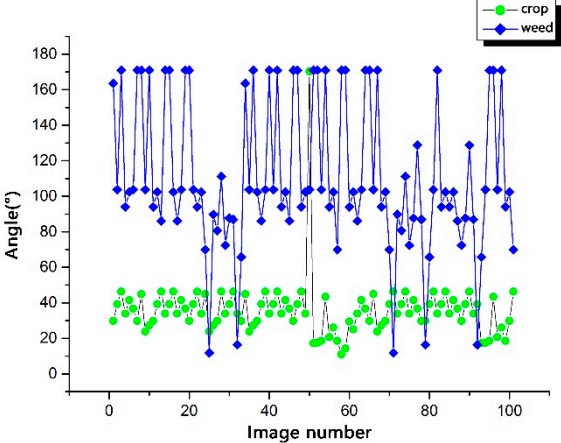

**Figure 13.** Angular characteristics of crop rows and weeds.

**Table 1.** Absolute corner information based on the row.

| Category | | Four Sets of Absolute Corner Coordinate Data | | | |
|---|---|---|---|---|---|
| Line | | First | Second | Third | Fourth |
| $C_{ul}(x,y)$ | $x_{ul}$ | 538 | 1070 | 1688 | 2120 |
| | $y_{ul}$ | 4 | 15 | 15 | 29 |
| $C_{ur}(x,y)$ | $x_{ur}$ | 714 | 1316 | 1896 | 2372 |
| | $y_{ur}$ | 5 | 27 | 33 | 45 |
| $C_{ll}(x,y)$ | $x_{ll}$ | 189 | 968 | 1818 | 2442 |
| | $y_{ll}$ | 570 | 564 | 554 | 568 |
| $C_{lr}(x,y)$ | $x_{lr}$ | 452 | 1248 | 2104 | 2738 |
| | $y_{lr}$ | 566 | 568 | 550 | 560 |
| | $s$ | —— | 787.5 | 853 | 629 |
| | $L_{wide}$ | 263 | 280 | 286 | 296 |
| | $L_{length}$ | 570.1618 | 572.2526 | 562.354 | 572.4229 |
| | $L_{aw}$ | 0.225975 | 0.240582 | 0.245737 | 0.254329 |
| | $L_{al}$ | 0.489894 | 0.491691 | 0.483186 | 0.491837 |

*3.3. Weed Density Parameters and Position*

The same kind of absolute corners to form a quadrilateral were marked and joined, as shown in Figure 14. In Figure 14a, the area inside the red quadrilateral was the crop row area and the rest was the inter-row area. The white pixels in the inter-row area represented weeds and white pixels in the crop row area represented corn crops. In Figure 14b, the crop row areas were labeled in red color.

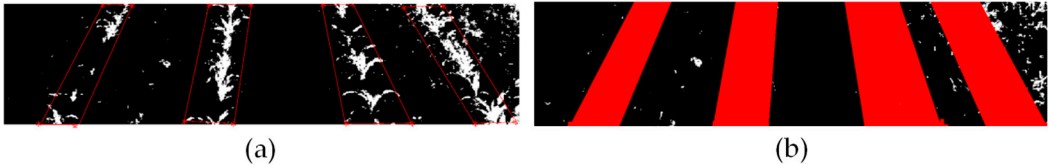

(a)  (b)

**Figure 14.** Absolute corner planning extract crop row: (**a**) Connected absolute corners; (**b**) Crop row area marker.

Finally, weed density parameters were calculated using the formulas in Section 2.3.2, the result data is shown in Table 2.

This study eliminated the crop row area and the remaining white pixel part was considered as weed area, as shown in Figure 15. Figure 15a shows the binary image after eliminating labeled crop row. Figure 15b presents the extracted and marked weed region and Figure 15c shows the position information of the detected weed.

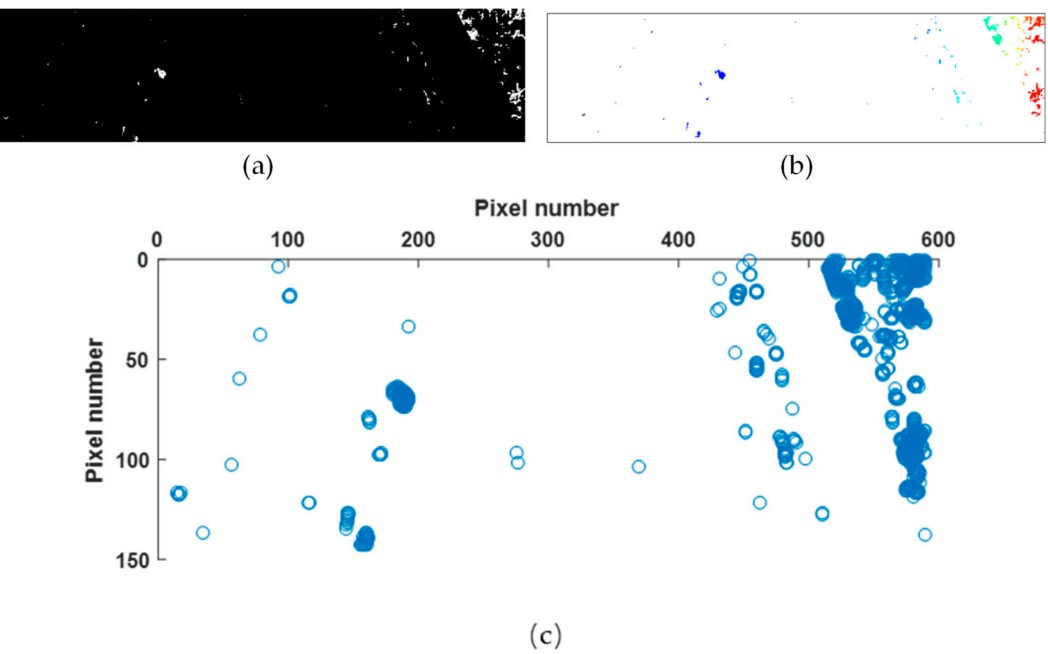

**Figure 15.** Crop row elimination and weed position extraction: (**a**) Crop row elimination; (**b**) Weed area extraction; (**c**) Weed position information.

### 3.4. Field Validation

A spraying experiment was carried out in the corn field and the driving speed of the spraying machine was set at 0.6 m/s.

In order to verify weed identification accuracy and speed, 50 weed images in the same period were processed by the algorithm we proposed and four traditional recognition algorithms, respectively. In order to compare the processing results, the identified images were labeled with grid, as shown in Figure 16. In grid image, one cell was marked as 1, half-cell was marked as 0.5, and two thirds cells were marked as 0.7. The number of squares occupied by the crop row area and weed area in the grid image were counted, then the weed area and the crop row area could be obtained. *Nw* represented inter-row weed pixel number in theory and *Nww* was the inter-row weed pixel number using identification methods; *Ncw* represented misidentification inter-row weed pixel number and *Nc* was the non-weed pixel number. The correct recognition rate of inter-row weeds $P_{CCR}$ and inter-row weed error recognition rate $P_{MCR}$ [35] are shown as follows:

$$\begin{cases} P_{CCR} = \frac{N_{WW}}{N_W} \times 100\% \\ P_{MCR} = \frac{N_{CW}}{N_C} \times 100\% \end{cases} \tag{32}$$

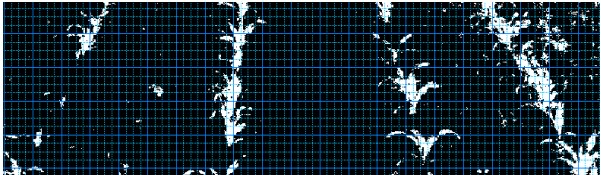

**Figure 16.** Gridded image for corn crops and weeds.

Table 3 showed the results for same weed images with size 2748 × 576 using the proposed algorithm and four traditional recognition methods, respectively. The computer was configured with Intel Core (TM) i5, 3.1 GHz, and 4 G memory.

**Table 3.** Recognition rates of the four algorithms for inter-row weeds.

| Recognition Algorithm | Correct Rate/% | Error Rate/% | Time/ms |
|---|---|---|---|
| Paper method | 90.3 | 4.7 | 782 |
| DBW algorithm | 78.5 | 5.3 | 1625 |
| Reference [2] algorithm | 91.2 | 1.6 | 2980 |
| Plant heart color | 68.6 | 11.2 | 550 |
| Based online width algorithm | 88.1 | 3.7 | 1483 |

The study also observed and recorded the liquid deposition of weed and maize; the results showed that the rate of weed application was 93.7%, maize application was 6.5%, and non-weed and non-maize application was 0.8%.

## 4. Discussion

### 4.1. Identification Accuracy and Speed Analysis

The identification accuracy and speed were of great significance, since they determined the possibility of the algorithm being operated in the real-time variable spraying system. The algorithm based on absolute corner did not need to directly identify and fit the center line of crop rows, which reduced a lot of algorithm load and saved algorithm processing time. The advantage over the recognition method in reference [8] was that the algorithm in this paper did not need to measure the row width of crops in different growth stages many times.

The results in Table 3 showed that the correct recognition rate of this study was 90.3% and the error recognition rate was 4.7%. In terms of speed, the method in this paper took 782 ms for all the processing steps, including 97 ms for grayscale conversion, 74 ms for binarization, 283 ms for corner detection, 269 ms for sub-corner classification, and 47 ms for absolute corner classification. Among them, the algorithm with highest accuracy was the method reference [2] (exploiting affine invariant regions and leaf edge shapes), yet this method had the highest time consumption. The plant heart color method took the least time (550 ms) but its correct identification rate was the lowest at 68.6%.

The accuracy of the proposed method varied with the changes of position between weeds and crops, i.e., the misidentification rate increases under the condition of irregular sowing and severer broken seedlings. From the above analysis, for the high-resolution image with 2748 × 576 pixels, the proposed algorithm took about 782 ms. For low-resolution images, the advantage in processing time of this method would be more significant and it can fully meet the real-time requirements in the field. For example, the time for processing an image with 600 × 800 pixels was 196 ms, which can ensure the field spaying is completed efficiently.

Such results showed the advantages of the proposed method in accuracy and speed aspects. Meanwhile, it can not only be applied to the identification of corn crop but also could be extended to other crops. In the future, additional field tests will be completed in other crops to verify the accuracy and usefulness of this method.

*4.2. Spraying Herbicide Analysis*

From the results of the liquid deposition of weed and maize in Section 3.4, the rate of weed application was 93.7%, maize application was 6.5%, and non-weed and non-maize application was 0.8%. Through observation and analysis, the reasons for these situations mainly include: some weeds were too close to the maize, the herbicide sprayed on the maize leaf, and the electromagnetic valve had a certain response time, causing the delay of spraying.

Currently, in China, excessive use of herbicides leads to serious effects, such as environmental and agricultural products pollution [36]. Compared to conventional spraying, this method can significantly reduce herbicide use and have great economic benefit and social benefit, which could be helpful in minimal farmland pollution, safety of agricultural products and personnel, and sustainable development of agriculture.

In conclusion, the proposed weed identification method was precise and rapid, which can effectively improve the weed identification efficiency and reduce the waste of herbicide as well as improve the herbicide utilization rate. Further, the method this study proposed was favorable for real-time field work, which provided the basis for precision agriculture development.

## 5. Conclusions

In this study, a weed density detection method based on AFPC algorithm was developed to guide weed management in field. The key conclusions of the present study were as follows:

(1) An AFPC extraction algorithm capable of detecting weed and crop corners was developed, with high detection accuracy and small computation load. In order to improve the processing speed of the AFPC algorithm, this study selected the corner distance as the threshold to filter Harris corners, which reduced processing time.

(2) A sub-corner classifier and an absolute corner classifier were developed to extract the sub-absolute corners and absolute corners. The angle threshold capable of eliminating pseudo-absolute corners was set according to the angle feature of weeds and crop rows. The absolute corners were merged to recognize the weed position without extracting the crop rows directly.

(3) Two weed density parameters were developed to calculate the weed pressure and weed cluster rate. Based on this, the weed distribution condition was evaluated for the entire farmland.

(4) The weed density detection method based on AFPC algorithm was validated in a corn field. Experiment results showed that the method was rapid and accurate, with processing time of 782 ms for an image of 2748 × 576 pixels and the correct rate for identifying weeds was up to 90.3%. In other words, the method this study developed could meet the real-time process requirement and could be used in actual production work.

**Author Contributions:** Conceptualization, Y.X. and R.H.; Methodology, Y.X. and Z.G.; Validation, R.H., Z.G. and C.L.; Writing-Original Draft Preparation, Y.Z.; Writing-Review & Editing, Z.G. and Y.Z.; Supervision, Y.J. and C.L.; Funding Acquisition, Y.X. All authors have read and agreed to the published version of the manuscript.

**Funding:** This research was funded by [National Natural Science Foundation of China] grant number [31801753] and [Jilin province education department "13 th five-year" science and technology research planning project] grant number [JJKH20200336KJ].

**Acknowledgments:** The authors wish to thank Q.Z., X.M., X.W., D.F. who help to make indoor and field experiments.

**Conflicts of Interest:** The authors declare no conflict of interest.

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
