# Peer review of "Weed Density Detection Method Based on Absolute Feature Corner Points in Field"

_agronomy, doi:10.3390/agronomy10010113_

Round 1
Reviewer 1 Report
Dear Authors
The method proposed in this paper aims at detecting crop rows. It is based on Harris corners. This approach is original. However, some aspects could be improved.
First, some equations and figures should be corrected/improved
Second, the naming of the different types of points/corners is confusing (absolute corners, sub-absolute corners, sub_contrast_corners, sub centroids...)
Third, please give us some information about the implementation. Indeed, the comparison of algorithms (in terms of computation time) is valid only if all the algorithms have been implemented on the same environment.
Here are my delailed comments
introduction
75-76 : position histogram [...] real-time farmland
84-84 : "calculation burden is relatively heavv"
Performance is not necessarily due to the method but to its implementation. Please give us information about the implementation.
section 2.1.1
127-127 : JiLin Agricultural University : add Changchun, China
131-131 : "according to the position". Please give us some details : depth zone, angle view...
section 2.1.2
146-158 : Using median filter is possible, but generally Wienner filter produce better results, and the noise strength should be studied.
equation (1) : missing parenthesis after min(Wn[Xij]
equation (2) : usually the median is the signal not the noise (I suppose "Noise" and "Signal" should be switched)
section 2.1.3
reference 17 : the section 2.1.3 presents the usual Otsu method, not the cited version (Otsu with ELM algorithm)
equation (3) the "t" variable in the should be (t-1) ? and "t" should be bounded (where t <= T-1)
equation (4) the "i" variable is duplicated. I propose to use "\sum{j=i+1}{T-1}{p_j}" (where i>=0) here "i" refer to equation (3)
Equation (5) is the same as (3)
Equation (6) is the same as (4)
section 2.1.4
179-179 : smaller weeds were removed using morphological optimization
What is the kernel used for that operation ? square ? circle ? Please mention its size.
190-194 : equation (10) erosion is the inverse operation of the dilatation. Therefore it is not a maximum, but a minimum. Perhaps bw(x,y) should be "+oo" outside the definition domain (unless it does not exist).
I wonder if -oo and +oo are the best choices for points outside the definition domain. In my opinion, excluding these points could profide better results
section 2.2.1
Equation (11) u,v is bounded to [-n/2 , n/2] where n is the window size. In addition the second term in the sum is incorrect (... + v \frac{\partial I}{\partial Y} + ...) (not X).
Please specify the approximations are based on taylor expansion
Equation (13) the term o(sqrt(u²+v²)) is lost by the use of the new definition of Wxy = exp(-(x²+y²)/d²). The transition is not explained
220-220 : In my opinion, the term C should not be squared (please verify that point)
221-221 : What is a real-time symmetric matrix ? It seems being a usual symmetric matrix.
section 2.2.2
230-247 : two parameter can also be used to influence the number of detected corners : the previously mentioned (k) and the windows size. In the paper k is fixed empirically to 0.04 and the windows size is fixed to 3x3. You should discuss this choice :
<< with a bigger k, you will get less false corners but you will also miss more real corners (high precision),
with a smaller k you will get a lot more corners, so you will miss less true corners, but get a lot of false ones (high recall). >>
section 2.2.3
251-255 : I do not understand the figure 3 and the associated comments. Please enhance this paragraph.
257-263 : if possible, please discuss the influence of the infestation level.
section 2.2.3.1
268-268 : "minimum positive enclosing rectangle" : prefer the generic term "minimum bounding box"
267 : I would prefer "centroid, bounding box (corners), Harris corners" instead "sub-contrast-corners, sub-centroid, sub-corner"
Perhaps you could show sub-absolute corners (relative corners ?) on figure 5
section 2.2.3.2
329-329 : "k" start at 0 in this sum notation
section 2.3.2
370-372 : what is the precision of the detected row-pitch ?
385-385 : duplicated definition of the term "i" in the equation. Please use i and j
380 : please define the "superior region"
General comment : this part is really difficult to understand (in my opinion because of the names chosen for the points/corners)
section 3.1
424-430 : you could compute statistics on connected components instead of removing elements with less than 15px areas
I wonder if figure 12a and 12b are not inverted... Please verify.
Section 3.2
Figure 11 and 12b are unreadable. Enhance contrast
section 3.4
table (3) : please give us information about the implementation.
Best regards
Author Response
Review Response: agronomy-650626
We thank the reviewers for taking the time to provide feedback on our manuscript. After reading the reviewer comments, we have carefully modified the manuscript to improve the work based on their suggestions. We provide a detailed explanation to address the reviewers’ specific comments.
Response to Reviewer 1 Comments
Introduction
75-76: position histogram [...] real-time farmland
84-84: "calculation burden is relatively heavy"
Performance is not necessarily due to the method but to its implementation. Please give us information about the implementation.
Response: Accepted. We have explained these in introduction part with yellow highlight.
Section 2.1.1
127-127: JiLin Agricultural University: add Changchun, China
131-131: "according to the position". Please give us some details: depth zone, angle view
Response: Accepted. We have added the “Changchun, China” in section 2.1.2 with yellow highlight.
About “according to the position”, we provided explanation in section 2.1.2 with yellow highlight.
Section 2.1.2
146-158: Using median filter is possible, but generally Wiener filter produce better results, and the noise strength should be studied.
Response: Explained. In this study, we used extremum median filter, this method can effectively eliminate random noise caused by weather or complex circumstances.
equation (1): missing parenthesis after min (Wn[Xij]
Response: Accepted. The missing parenthesis in equation (1) was added with yellow highlight.
equation (2): usually the median is the signal not the noise (I suppose "Noise" and "Signal" should be switched)
Response: Accepted. We verified this equation, it is correct.
Section 2.1.3
reference 17: the section 2.1.3 presents the usual Otsu method, not the cited version (Otsu with ELM algorithm)
Response: Explained. In reference 17, it has the content about the usual Otsu, which was cited in this study.
equation (3) the "t" variable in the should be (t-1)? and "t" should be bounded (where t <= T-1)
Response: Explained. “t” is the threshold value, and equation (3) counted the grey value from 0 to t. So, this equation is correct.
equation (4) the "i" variable is duplicated. I propose to use "\sum{j=i+1} {T-1} {p_j}" (where i>=0) here "i" refer to equation (3)
Response: Explained. We checked equation (4), the value of “i” should begin with t+1. It has been modified in equation (4) with yellow highlight.
Equation (5) is the same as (3)
Response: Explained. “t” is the threshold value, and equation (3) counted the grey value from 0 to t. So, this equation is correct.
Equation (6) is the same as (4)
Response: Explained. We checked the equations, the value of “i” should begin with t+1. It has been modified in equation (6).
Section 2.1.4
179-179: smaller weeds were removed using morphological optimization
What is the kernel used for that operation? square? circle? Please mention its size.
Response: Accepted. We provided the kernel in Line: 181 with yellow highlight.
190-194: equation (10) erosion is the inverse operation of the dilatation. Therefore, it is not a maximum, but a minimum. Perhaps bw(x,y) should be "+oo" outside the definition domain (unless it does not exist).
I wonder if -oo and +oo are the best choices for points outside the definition domain. In my opinion, excluding these points could provide better results
Response: Accepted and Explained. Equation (10) was modified with yellow highlight. About the -oo, it has little effect, so it is an assumed value.
Section 2.2.1
Equation (11) u,v is bounded to [-n/2 , n/2] where n is the window size. In addition the second term in the sum is incorrect (... + v \frac {\partial I} {\partial Y} + ...) (not X).
Please specify the approximations are based on Taylor expansion
Response: Accepted. Equation (11) was modified and “based on Taylor expansion “with yellow highlight was added.
Equation (13) the term o(sqrt(u²+v²)) is lost by the use of the new definition of Wxy = exp(-(x²+y²)/d²). The transition is not explained
Response: Accepted. Equation (13) has been modified with yellow highlight according to the review.
220-220: In my opinion, the term C should not be squared (please verify that point)
Response: Accepted. We verified the expression for term C which is correct.
221-221: What is a real-time symmetric matrix? It seems being a usual symmetric matrix.
Response: Explained. M is a real symmetric matrix and not a real-time symmetric matrix, which was modified with yellow highlight.
Section 2.2.2
230-247: two parameters can also be used to influence the number of detected corners: the previously mentioned (k) and the windows size. In the paper k is fixed empirically to 0.04 and the windows size is fixed to 3×3. You should discuss this choice
Response: Explained. We provided detailed explanation in section 2.2.1 and 2.2.2 with yellow highlight.
Section 2.2.3
251-255: I do not understand the figure 3 and the associated comments. Please enhance this paragraph.
Response: Explained. We modified this explanation in Lines: 258-260 of section 2.2.2 with yellow highlight.
257-263: if possible, please discuss the influence of the infestation level.
Response: Thank you so much for your suggestion. Since the infestation level has less relation with the content. So, the authors prefer to without adding such content.
Section 2.2.3.1
268-268: "minimum positive enclosing rectangle": prefer the generic term "minimum bounding box"
Response: Accepted. The content has been modified in section 2.2.3.1 with yellow highlight based on reviewer’s suggestion.
267: I would prefer "centroid, bounding box (corners), Harris corners" instead "sub-contrast-corners, sub-centroid, sub-corner"
Response: Accepted. The corresponded content has been modified.
Perhaps you could show sub-absolute corners (relative corners?) on figure 5
Response: Explained. Thank you so much for the suggestion. The sub-absolute is the intermediate result, which is not suitable to show in figure 5.
Section 2.2.3.2
329-329: "k" starts at 0 in this sum notation
Response: Explained. The k represents the number of sub-centroids, so it started at 1.
Section 2.3.2
370-372: what is the precision of the detected row-pitch?
Response: Explained. This study didn’t contain the precision of the detected row-pitch, while it included total recognition precision, as is shown in Section 3.4 and 4.1.
385-385: duplicated definition of the term "i" in the equation. Please use i and j
Response: Accepted. We modified i in Line: 387 according to the review with yellow highlight.
380: please define the "superior region"
Response: Explained. The “superior region” is the “optimal region”, that was defined in section 2.1.1. We have modified “superior region” to “optimal region” with yellow highlight.
General comment: this part is really difficult to understand (in my opinion because of the names chosen for the points/corners)
Response: Explained. About this point, we used abbreviations for some commonly used noun.
Sub-corner --- SCO Sub-contrast corner --- SCC
Sub-centroid --- SCE Sub-absolute corner --- SAC
absolute corner --- AC absolute centroid --- AE
All of these were shown in 2.2.3 with yellow highlight.
Section 3.1
424-430: you could compute statistics on connected components instead of removing elements with less than 15 px areas
Response: Explained. The experiments results found that the area less than 15 px were not important to the following processing, so such method was applied in our study.
I wonder if figure 12 a and 12 b are not inverted... Please verify.
Response: Explained. We verified the figure 12 a and 12 b are correct.
Section 3.2
Figure 11 and 12 b are unreadable. Enhance contrast
Response: Accepted. Such two figures have been modified.
Section 3.4
table (3): please give us information about the implementation.
Response: Explained. the implementation was provided in section 3.4 with yellow highlight.
Reviewer 2 Report
This was an interesting study, but the methodology, results and conclusions are difficult to decipher due to poor organization and English.
Overall, this paper is worthy of publication but I would suggest the authors have someone review the manuscript in its entirety just for proofing the English and verb tense. All throughout the paper, the authors switched between current, future, and past tense.
I would also suggest that the formulas that are presented for the alogrithms be written on separate lines individually as they were merged with regular text in some instances and hard to follow. I also feel that the introduction is a bit choppy - the results of previous studies are stated in sentences that do not flow together well.
I have noted several comments and edits in the attached file, but these are not all encompassing of all the errors that I found. Please review the paper and have someone review it for clarity and presentation and resubmit.

Author Response
Review Response: agronomy-650626
We thank the reviewers for taking the time to provide feedback on our manuscript. After reading the reviewer comments, we have carefully modified the manuscript to improve the work based on their suggestions. We provide a detailed explanation to address the reviewers’ specific comments.
Response to Reviewer 1 Comments
Introduction
75-76: position histogram [...] real-time farmland
84-84: "calculation burden is relatively heavy"
Performance is not necessarily due to the method but to its implementation. Please give us information about the implementation.
Response: Accepted. We have explained these in introduction part with yellow highlight.
Section 2.1.1
127-127: JiLin Agricultural University: add Changchun, China
131-131: "according to the position". Please give us some details: depth zone, angle view
Response: Accepted. We have added the “Changchun, China” in section 2.1.2 with yellow highlight.
About “according to the position”, we provided explanation in section 2.1.2 with yellow highlight.
Section 2.1.2
146-158: Using median filter is possible, but generally Wiener filter produce better results, and the noise strength should be studied.
Response: Explained. In this study, we used extremum median filter, this method can effectively eliminate random noise caused by weather or complex circumstances.
equation (1): missing parenthesis after min (Wn[Xij]
Response: Accepted. The missing parenthesis in equation (1) was added with yellow highlight.
equation (2): usually the median is the signal not the noise (I suppose "Noise" and "Signal" should be switched)
Response: Accepted. We verified this equation, it is correct.
Section 2.1.3
reference 17: the section 2.1.3 presents the usual Otsu method, not the cited version (Otsu with ELM algorithm)
Response: Explained. In reference 17, it has the content about the usual Otsu, which was cited in this study.
equation (3) the "t" variable in the should be (t-1)? and "t" should be bounded (where t <= T-1)
Response: Explained. “t” is the threshold value, and equation (3) counted the grey value from 0 to t. So, this equation is correct.
equation (4) the "i" variable is duplicated. I propose to use "\sum{j=i+1} {T-1} {p_j}" (where i>=0) here "i" refer to equation (3)
Response: Explained. We checked equation (4), the value of “i” should begin with t+1. It has been modified in equation (4) with yellow highlight.
Equation (5) is the same as (3)
Response: Explained. “t” is the threshold value, and equation (3) counted the grey value from 0 to t. So, this equation is correct.
Equation (6) is the same as (4)
Response: Explained. We checked the equations, the value of “i” should begin with t+1. It has been modified in equation (6).
Section 2.1.4
179-179: smaller weeds were removed using morphological optimization
What is the kernel used for that operation? square? circle? Please mention its size.
Response: Accepted. We provided the kernel in Line: 181 with yellow highlight.
190-194: equation (10) erosion is the inverse operation of the dilatation. Therefore, it is not a maximum, but a minimum. Perhaps bw(x,y) should be "+oo" outside the definition domain (unless it does not exist).
I wonder if -oo and +oo are the best choices for points outside the definition domain. In my opinion, excluding these points could provide better results
Response: Accepted and Explained. Equation (10) was modified with yellow highlight. About the -oo, it has little effect, so it is an assumed value.
Section 2.2.1
Equation (11) u,v is bounded to [-n/2 , n/2] where n is the window size. In addition the second term in the sum is incorrect (... + v \frac {\partial I} {\partial Y} + ...) (not X).
Please specify the approximations are based on Taylor expansion
Response: Accepted. Equation (11) was modified and “based on Taylor expansion “with yellow highlight was added.
Equation (13) the term o(sqrt(u²+v²)) is lost by the use of the new definition of Wxy = exp(-(x²+y²)/d²). The transition is not explained
Response: Accepted. Equation (13) has been modified with yellow highlight according to the review.
220-220: In my opinion, the term C should not be squared (please verify that point)
Response: Accepted. We verified the expression for term C which is correct.
221-221: What is a real-time symmetric matrix? It seems being a usual symmetric matrix.
Response: Explained. M is a real symmetric matrix and not a real-time symmetric matrix, which was modified with yellow highlight.
Section 2.2.2
230-247: two parameters can also be used to influence the number of detected corners: the previously mentioned (k) and the windows size. In the paper k is fixed empirically to 0.04 and the windows size is fixed to 3×3. You should discuss this choice
Response: Explained. We provided detailed explanation in section 2.2.1 and 2.2.2 with yellow highlight.
Section 2.2.3
251-255: I do not understand the figure 3 and the associated comments. Please enhance this paragraph.
Response: Explained. We modified this explanation in Lines: 258-260 of section 2.2.2 with yellow highlight.
257-263: if possible, please discuss the influence of the infestation level.
Response: Thank you so much for your suggestion. Since the infestation level has less relation with the content. So, the authors prefer to without adding such content.
Section 2.2.3.1
268-268: "minimum positive enclosing rectangle": prefer the generic term "minimum bounding box"
Response: Accepted. The content has been modified in section 2.2.3.1 with yellow highlight based on reviewer’s suggestion.
267: I would prefer "centroid, bounding box (corners), Harris corners" instead "sub-contrast-corners, sub-centroid, sub-corner"
Response: Accepted. The corresponded content has been modified.
Perhaps you could show sub-absolute corners (relative corners?) on figure 5
Response: Explained. Thank you so much for the suggestion. The sub-absolute is the intermediate result, which is not suitable to show in figure 5.
Section 2.2.3.2
329-329: "k" starts at 0 in this sum notation
Response: Explained. The k represents the number of sub-centroids, so it started at 1.
Section 2.3.2
370-372: what is the precision of the detected row-pitch?
Response: Explained. This study didn’t contain the precision of the detected row-pitch, while it included total recognition precision, as is shown in Section 3.4 and 4.1.
385-385: duplicated definition of the term "i" in the equation. Please use i and j
Response: Accepted. We modified i in Line: 387 according to the review with yellow highlight.
380: please define the "superior region"
Response: Explained. The “superior region” is the “optimal region”, that was defined in section 2.1.1. We have modified “superior region” to “optimal region” with yellow highlight.
General comment: this part is really difficult to understand (in my opinion because of the names chosen for the points/corners)
Response: Explained. About this point, we used abbreviations for some commonly used noun.
Sub-corner --- SCO Sub-contrast corner --- SCC
Sub-centroid --- SCE Sub-absolute corner --- SAC
absolute corner --- AC absolute centroid --- AE
All of these were shown in 2.2.3 with yellow highlight.
Section 3.1
424-430: you could compute statistics on connected components instead of removing elements with less than 15 px areas
Response: Explained. The experiments results found that the area less than 15 px were not important to the following processing, so such method was applied in our study.
I wonder if figure 12 a and 12 b are not inverted... Please verify.
Response: Explained. We verified the figure 12 a and 12 b are correct.
Section 3.2
Figure 11 and 12 b are unreadable. Enhance contrast
Response: Accepted. Such two figures have been modified.
Section 3.4
table (3): please give us information about the implementation.
Response: Explained. the implementation was provided in section 3.4 with yellow highlight.
Response to Reviewer 2 Comments
This was an interesting study, but the methodology, results and conclusions are difficult to decipher due to poor organization and English.
Overall, this paper is worthy of publication, but I would suggest the authors have someone review the manuscript in its entirety just for proofing the English and verb tense. All throughout the paper, the authors switched between current, future, and past tense.
I would also suggest that the formulas that are presented for the algorithms be written on separate lines individually as they were merged with regular text in some instances and hard to follow. I also feel that the introduction is a bit choppy - the results of previous studies are stated in sentences that do not flow together well.
I have noted several comments and edits in the attached file, but these are not all encompassing of all the errors that I found. Please review the paper and have someone review it for clarity and presentation and resubmit.
Response: Thank you so much for your suggestions.
The manuscript has been reviewed by the native English-speaking colleague thoroughly, and the corresponded errors (especially about verb tense) have been modified.
The modified content in the manuscript according to reviewers 2 has been highlighted in red. Meanwhile, the formulas have been presented in separate lines, as a result, four more formulas were presented in this manuscript highlighted in yellow.
